# TapMo: Shape-aware Motion Generation of Skeleton-free Characters

**Jiaxu Zhang**[1,2*]**, Shaoli Huang**[2†]**, Zhigang Tu**[1‡]**,**
**Xin Chen**[3]**, Xiaohang Zhan**[2]**, Gang Yu**[3]**, Ying Shan**[2]

[1]Wuhan University, [2]Tencent AI Lab, [3]Tencent PCG

## Abstract

Previous motion generation methods are limited to the pre-rigged 3D human model, hindering their applications in the animation of various non-rigged characters. In this work, we present TapMo, a **T**ext-driven **A**nimation **P**ipeline for synthesizing **Mo**tion in a broad spectrum of skeleton-free 3D characters. The pivotal innovation in TapMo is its use of shape deformation-aware features as a condition to guide the diffusion model, thereby enabling the generation of mesh-specific motions for various characters. Specifically, TapMo comprises two main components - Mesh Handle Predictor and Shape-aware Diffusion Module. Mesh Handle Predictor predicts the skinning weights and clusters mesh vertices into adaptive handles for deformation control, which eliminates the need for traditional skeletal rigging. Shape-aware Motion Diffusion synthesizes motion with mesh-specific adaptations. This module employs text-guided motions and mesh features extracted during the first stage, preserving the geometric integrity of the animations by accounting for the character's shape and deformation. Trained in a weakly-supervised manner, TapMo can accommodate a multitude of non-human meshes, both with and without associated text motions. We demonstrate the effectiveness and generalizability of TapMo through rigorous qualitative and quantitative experiments. Our results reveal that TapMo consistently outperforms existing auto-animation methods, delivering superior-quality animations for both seen or unseen heterogeneous 3D characters. The project page: https://semanticdh.github.io/TapMo.

## 1 Introduction

The advent of 3D animation has revolutionized the digital world, providing a vibrant platform for storytelling and visual design. However, the intricacy and time-consuming nature of traditional animation processes poses significant barriers to entry. In recent years, learning-based methods have emerged as a promising solution for animation. Motion generation models (Guo et al., 2020; Zhang et al., 2022) and auto-rigging methods (Xu et al., 2020; 2022b) are among the techniques that offer unprecedented results for motion synthesis and mesh control, respectively, in both quality and generalization. Nevertheless, they still fall short of providing a comprehensive solution.

Existing motion generation methods (Tevet et al., 2022; Zhang et al., 2022; Chen et al., 2023) facilitate animation by leveraging the SMPL model (Loper et al., 2015), a parametric 3D model of the human form equipped with a uniform skeletal rigging system. However, these approaches oversimplify the task by assuming a fixed mesh topology and skeletal structures, and they fail to account for the geometric shapes of skeleton-free characters. This inherent assumption significantly restricts the creation of motions for diverse, non-rigged 3D characters. Moreover, the predominant focus of existing motion datasets is on humanoid characters, which consequently limits the potential for data-driven methods to be applicable to non-standard characters. Therefore, a significant challenge lies in extending motion generation models from human to non-human characters, enabling the model to accurately perceive their geometry even in the absence of pre-rigged skeletons.

---

[*]Most of this work was done during Jiaxu's internship at Tencent AI Lab. Email: zjiaxu@whu.edu.cn
[†]Jiaxu Zhang and Shaoli Huang contributed equally.
[‡]Corresponding author: tuzhigang@whu.edu.cn

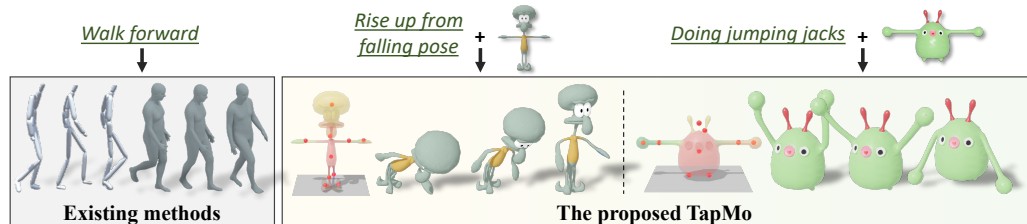

Figure 1: We present TapMo, a text-based animation pipeline for generating motion in a wide variety of skeleton-free characters.

On the other hand, auto-rigging methods have attempted to enable mesh deformation control by skeletonizing and rigging 3D characters using optimization techniques (Baran & Popović, 2007) or deep models (Xu et al., 2020; 2019). Despite their attempts, these methods tend to generate diverse skeleton structures for different characters, hindering the possibility of generating a uniform motion representation for animation. It is worth highlighting that certain methods have suggested the utilization of a pre-defined skeleton template for rigging characters (Liu et al., 2019b; Li et al., 2021). However, these approaches are incapable of handling meshes that exhibit substantial variations in shape and topology. Consequently, the resulting rigging often necessitates manual adjustments or custom motions, which undermines the objective of achieving automatic motion synthesis.

To overcome these limitations, we introduce a new Text-driven Animation Pipeline (TapMo) capable of generating realistic and anthropomorphic motion for a wide range of skeleton-free 3D characters as Figure 1 shows. TapMo only requires the input of motion descriptions in natural language and can animate skeleton-free characters by perceiving their shapes and generating mesh-specific motions. To achieve this goal, we explore two key components of TapMo, namely the *Mesh Handle Predictor* and the *Shape-aware Motion Diffusion*. These two components synergistically automate the entire animation creation process, establishing a more accessible way of generating animations beyond SMPL-based humans, so as to democratize animation creation on heterogeneous characters.

The Mesh Handle Predictor addresses the mesh deformation control in a unified manner for diverse 3D characters. Inspired by SfPT (Liao et al., 2022), we utilize a GCN-based network that adaptively locates control handles in the mesh, regulating their movements based on predicted skinning weights. Each predicted handle independently controls its assigned semantic part, while the first handle is fixed as the character's root for global control. More importantly, the module also serves a dual purpose by offering a mesh deformation feature for future shape-aware motion generation.

The Shape-aware Motion Diffusion copes with the challenge of perceiving the mesh shape and generating realistic motions for heterogeneous 3D characters. Language descriptors are regarded as a user-friendly interface for people to interact with computers (Manaris, 1998). Thus, we design a text-driven motion generation model with respect to the Diffusion Model (Sohl-Dickstein et al., 2015; Ho et al., 2020) that has achieved impressive results in the generation of complex data distribution recently. Different from previous Motion Diffusion Models (Tevet et al., 2022; Zhang et al., 2022), our model generates a universal motion representation that does not rely on the character's skeletal rigging structure, enabling it to animate a wide range of characters. Notably, we use the mesh deformation feature encoded by the Mesh Handle Predictor as an additional condition to generate motion adaptations that are flexible for the specific mesh to preserve the geometric integrity of the animations. Finally, Linear Blend Skinning is used to animate the mesh with the generated motion and the predicted handles.

In response to the challenge posed by the absence of ground-truth handles and motions for diverse 3D characters, we present a novel, weakly-supervised training approach for TapMo. Our strategy leverages shape constraints and motion priors as auxiliary supervision signals to enhance the learning of handle prediction and motion generation in TapMo.

We evaluate our TapMo on a variety of 3D characters and complex motion descriptions. The qualitative and quantitative results show that our TapMo is effective on seen and unseen 3D characters. The generated animation of TapMo achieves higher motion and geometry quality compared with existing learning-based methods.

Our contributions are listed below:

- We present a comprehensive solution TapMo that, to our knowledge, represents the first attempt to enable the motion generation of generic skeleton-free characters using text descriptions.

- We design two key components in TapMo *i.e.*, the Mesh Handle Predictor and the Shape-aware Motion Diffusion. These components collaborate to govern mesh deformation and generate plausible motions while accounting for diverse non-rigged mesh shapes.

- A weakly-supervised training strategy is presented to enable the geometry learning of TapMo with limited ground-truth data of both handle-annotated characters and text-relevant motions.

- Extensive experiments demonstrate the effectiveness and generalizability of TapMo qualitatively and quantitatively, and the animation results of TapMo are superior to existing learning-based methods both on motion and geometry.

## 2 RELATED WORK

**Motion generation** remains a persistent challenge within the realm of computer graphics (Guo et al., 2015). Recently, an upsurge of interest has been seen in learning-based models for motion generation (Holden et al., 2015; 2016). These cutting-edge models often explore various conditions for human motion generation, which include but are not limited to action labels (Guo et al., 2020; Petrovich et al., 2021), audio (Xu et al., 2022a; Dabral et al., 2022), and text (Tevet et al., 2022; Chen et al., 2023; Zhang et al., 2022). Among all these conditioned modalities, text descriptors are identified as the most user-friendly and convenient, thereby positioning text-to-motion as a rising area of research. While the existing advancements pave the way for the motion generation of humans equipped with a pre-defined skeletal rigging system, they do not cater to various 3D characters lacking this skeletal system. Moreover, several large-scale datasets, providing human motions as sequences of 3D skeletons (Liu et al., 2019a; Guo et al., 2020; 2022) or SMPL parameters (Mahmood et al., 2019). However, these datasets also primarily cater to human motion learning and lack the inclusion of a broad spectrum of heterogeneous characters. Consequently, methods trained directly on these datasets struggle to adapt to the task of generating motions for heterogeneous characters. Acknowledging this limitation, our research sets sail into previously unexplored territory: text-driven motion generation for skeleton-free 3D characters. Distinctly different from previous studies, the 3D characters in our approach are non-rigged and feature a wide array of mesh topologies.

**Skeleton-free mesh deformation**. The skeletal rigging system is widely used in animation (Magnenat et al., 1988), but the process of skeletonization and rigging is not accessible to laypeople. To address this issue, some researchers have explored skeleton-free mesh deformation methods. Traditional methods rely on manual landmark annotations (Ben-Chen et al., 2009; Sumner & Popović, 2004). Recently, Yifan et al. (2020) proposed a learnable neural cage for detail-preserving shape deformation, which works well for rigid objects but not for articulated characters. Other approaches, such as those by Tan et al. (2018), Yang et al. (2021), embed the mesh into latent spaces to analyze the mesh deformation primitives. However, these latent spaces are not shared across subjects, so they cannot be used for animating different characters. Wang et al. (2023b) propose a zero-shot implicit deformation module, which is incompatible with our explicit motion generation approach. The recent SfPT (Liao et al., 2022) and HMC (Wang et al., 2023a) generate consistent deformation parts across different character meshes, allowing for pose transfer through the corresponding parts. However, the character's root is ignored in the deformation parts, leading to failure in motion transfer. In this work, we are inspired by SfPT and introduce a Mesh Handle Predictor to animate skeleton-free characters in a handle-based manner. This manner represents motion using the translation and rotation of predicted mesh handles, which differs from traditional skeleton-based methods. Our approach diverges from SfPT in two key respects: first, unlike SfPT, we incorporate root handle prediction to enable more intricate motions. Second, our handle predictor also functions to deliver a mesh deformation-aware representation that is crucial for subsequent mesh motion generation.

## 3 METHOD

As illustrated in Figure 2, given a non-rigged mesh and a motion description in natural language, our goal is to synthesize a realistic animation for the character according to the motion description $\kappa$. Specifically, the input mesh of the character $\phi$ is fed into the Mesh Handle Predictor $\lambda(\cdot)$ to predict the mesh handle $\boldsymbol{h}$ and the skinning weight $\boldsymbol{s}$ of the vertices. Besides, a low-dimensional mesh deformation feature $\boldsymbol{f}_\phi$ of the character $\phi$ is extracted to represent the shape of the mesh. Let $\boldsymbol{\theta}_\lambda$ denote the learnable parameter of the Mesh Handle Predictor, this process is formulated as:

$$\lambda : (\phi; \boldsymbol{\theta}_\lambda) \mapsto (\boldsymbol{h}, \boldsymbol{s}, \boldsymbol{f}_\phi). \tag{1}$$

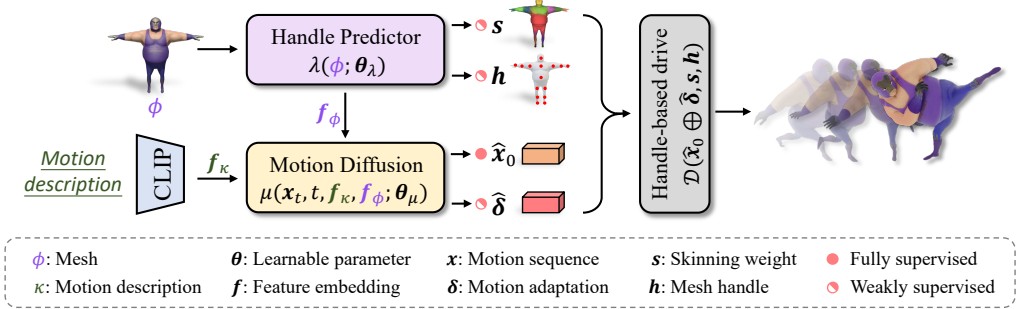

Figure 2: **An overview of the TapMo pipeline.** Given a non-rigged mesh and a motion description input by the user, the *Mesh Handle Predictor* $\lambda(\cdot)$ predicts mesh handles and skinning weights to control the mesh. The *Shape-aware Motion Diffusion* $\mu(\cdot)$ generates a text-guided and mesh-specific motion for the character using the motion description and the mesh deformation feature $\boldsymbol{f}_\phi$ extracted by the Mesh Handle Predictor.

Subsequently, in Shape-aware Motion Diffusion, the mesh-deformation feature $\boldsymbol{f}_\phi$ and the text embedding of the motion description $\boldsymbol{f}_\kappa$ are used as the conditions to generate a text-guided motion $\hat{\boldsymbol{x}}_0$ and a mesh-specific adaptation $\hat{\boldsymbol{\delta}}$. The text embedding $\boldsymbol{x}_\kappa$ is extracted by the CLIP (Radford et al., 2021) model. This inverse diffuse process is formulated as:

$$\mu : (\boldsymbol{x}_t, t, \boldsymbol{f}_\kappa, \boldsymbol{f}_\phi; \boldsymbol{\theta}_\mu) \mapsto (\hat{\boldsymbol{x}}_0, \hat{\boldsymbol{\delta}}), \qquad (2)$$

where $\mu(\cdot)$ is the conditional denoiser of Diffusion Model and $\boldsymbol{x}_t$ is the motion of noising step $t$. $\boldsymbol{\theta}_\mu$ is the learnable parameter of $\mu(\cdot)$. Following Tevet et al. (2022), we predict the real motion $\hat{\boldsymbol{x}}_0$, rather than the noise $\hat{\epsilon}_t$. Finally, after $T = 1,000$ times of the inverse diffuse process, the sum of the generated motion $\hat{\boldsymbol{x}}_0$ and the adaptation $\hat{\boldsymbol{\delta}}$ are applied to the character via a handle-based driving method $\mathcal{D}(\cdot)$ with the predicted $\boldsymbol{h}$ and $\boldsymbol{s}$.

## 3.1 MESH HANDLE PREDICTOR

Given a mesh $\phi$ consisting of $V$ vertices, we designate $K$ mesh handles $\{\boldsymbol{h}_k\}_{k=1}^K$ to govern its deformation. Consequently, every vertex is associated with a $K$-dimensional skinning weight associated with $K$ handles. Overall, for the mesh $\phi$, the skinning weight is defined as $\boldsymbol{s} \in \mathbb{R}^{V \times K}$, where $0 \le s_{i,k} \le 1$ and $\sum_{k=1}^K s_{i,k} = 1$. $i$ is the vertex index. Since characters may have varying shapes and topologies, each handle is dynamically assigned to vertices with the same semantics across different meshes, except that the first handle is fixed at the root of the mesh. This adaptive assignment approach enables effective control of local and global motion over heterogeneous 3D characters.

Based on the definition of the mesh handles, the motion of the mesh can be expressed as the combination of the translations and rotations of the handles, which involves local translations $\boldsymbol{\tau}^l \in \mathbb{R}^{(K-1) \times 3}$, local rotations $\boldsymbol{r}^l \in \mathbb{R}^{(K-1) \times 6}$, global translation $\boldsymbol{\tau}^g \in \mathbb{R}^3$, and global rotation $\boldsymbol{r}^g \in \mathbb{R}^3$. The rotations are represented by the 6D features proposed by Zhou et al. (2019). This motion can be denoted as $\boldsymbol{x}_0 = \{\boldsymbol{\tau}^l, \boldsymbol{r}^l, \boldsymbol{\tau}^g, \boldsymbol{r}^g\}$ and the mesh can be driven by the motion in a handle-based manner $\mathcal{D}(\cdot)$ according to the following formula:

$$\bar{\boldsymbol{V}}_i^{'} = \boldsymbol{r}^g \left( \sum_{k=2}^K \boldsymbol{s}_{i,k} \left( \boldsymbol{r}_k^l (\bar{V}_i - \boldsymbol{h}_k) + \boldsymbol{\tau}_k^l \right) + \boldsymbol{h}_k \right) + \boldsymbol{\tau}^g, \forall \bar{V}_i \in \bar{\boldsymbol{V}}, \qquad (3)$$

where $\bar{\boldsymbol{V}} \in \mathbb{R}^{V \times 3}$ is the positions of the mesh vertices. The rotations in this formula are represented by the rotation matrix.

In our TapMo, following Liao et al. (2022), we employ the Mesh Handle Predictor based on a GCN to locate the handles for the mesh. The input feature is the positions and normal of the mesh vertices, which can be represented as $\phi \in \mathbb{R}^{V \times 6}$. The outputs are the skinning weight of mesh vertices and the handle positions in terms of the average position of vertices weighted by the skinning weight.

**Adaptive Handle Learning**. The existing 3D character datasets (Xu et al., 2019; Adobe) lack handle annotations, and the skinning weights are defined independently for each character, which lacks consistency. Therefore, these datasets cannot be directly used as ground-truth. To overcome this limitation, we introduce three losses, *i.e.*, Skinning Loss $\mathcal{L}_s$, Pose Loss $\mathcal{L}_p$, and Root Loss $\mathcal{L}_r$

Figure 3: **The structure and the training strategy of Shape-aware Motion Diffusion.** The Motion Diffusion learns to generate shape-aware motions based on the textual descriptions, as well as the mesh deformation features with only the ground-truth text-motion data of the SMPL model. The parameters of the Handle Predictor are frozen during the second stage to provide stable mesh deformation features. $x^{sk}$ is a random motion sequence in skeletal representation.

to achieve an adaptive handle learning. Among them, the Skinning Loss $\mathcal{L}_s$ and Pose Loss $\mathcal{L}_p$ stem from Liao et al. (2022) and will be detailed in the Appendix.

Root Loss $\mathcal{L}_r$ is to enforce the first handle point close to the character's center hip, which is an important step in animation for controlling the global motion independently. Liao et al. (2022) do not take into account the global motion of the character and thus produce severe mesh distortion when applied to drive complex motions. We observe that in the traditional skeleton rigging system, the first joint of the character is always located in the hip part, which is also considered the root joint of the character. Thus, we use $\mathcal{L}_r$ to make the predicted position of the first handle $h_1$ close to the position of the root joint $b_1$ in the character. $\mathcal{L}_r$ is formulated as:

$$\mathcal{L}_r = \|h_1 - b_1\|_2^2$$
$$= \left\| \frac{\sum_{i=1}^{V} s_{i,1} \bar{V}_i}{\sum_{i=1}^{V} s_{i,1}} - b_1 \right\|_2^2. \tag{4}$$

With the three losses introduced above, the Handle Predictor can be trained by:

$$\min_{\boldsymbol{\theta}_\lambda} \ \mathcal{L}_s + \nu_p \mathcal{L}_p + \nu_r \mathcal{L}_r, \tag{5}$$

where $\nu_r$ and $\nu_p$ are the loss balancing factor.

## 3.2 SHAPE-AWARE MOTION DIFFUSION

The Mesh Handle Predictor allows for controlling heterogeneous characters using a unified motion representation described in Eq.3. However, directly generating this motion representation to synthesize animation can lead to significant mesh distortion due to large variations in body shape among different characters. To address this issue, the Shape-aware Motion Diffusion generates mesh-specific adaptation parameters in addition to the handle motion. This approach preserves the geometric integrity of the animations.

The left part of Figure 3 illustrates the structure of the Shape-aware Motion Diffusion. The model takes the noising step $t$, the motion description feature $\boldsymbol{f}_\kappa$, and the mesh deformation feature $\boldsymbol{f}_\phi$ as conditions while the $t$-step noised motion $\boldsymbol{x}_t$ as input to generate the text-guided motion $\hat{\boldsymbol{x}}_0^{1:N}$ of $N$ frames and the mesh-specific adaptation $\hat{\boldsymbol{\delta}}^{1:N}$ that are suitable for the mesh $\phi$. This process is defined in Eq.2. For brevity, and when there is no danger of confusion, we omit the superscripts.

We employ a Transformer Decoder (Vaswani et al., 2017) to construct the denoiser of the Diffusion Model and use two linear layers to map the output feature into motion and adaptation, respectively. Unlike MDM (Tevet et al., 2022), which uses only one text token from the last layer of CLIP, we use multiple text tokens from the penultimate layer of CLIP, the number of which is the same as the number of words in the input motion prompt. We have observed that the motion distortion is primarily caused by the generated local translations in our motion representation. Therefore, we only apply adaptations to the local translations of the generated motion. Thus, the dimension of the motion adaptation in one frame is $K \times 3$. The final local translations in the motion are obtained by summing up the generated local translations and the mesh-specific adaptations.

To train the Diffusion Model, we fully utilize the existing motion-language dataset (Guo et al., 2022) by converting SMPL motions to our handle-based motion representation using the analytical method

introduced in Besl & McKay (1992) as the ground-truth motion $\boldsymbol{x}_0$. Originating from the simple loss of DDPM (Ho et al., 2020), a fully-supervised Motion Loss $\mathcal{L}_m$ is formulated as:

$$\mathcal{L}_m := \mathbb{E}_{\boldsymbol{x}_0, t, \boldsymbol{f}_\kappa, \boldsymbol{f}_\phi} \left[ \|\boldsymbol{x}_0 - \hat{\boldsymbol{x}}_0\|_2^2 \right], \tag{6}$$

where the motion $\hat{\boldsymbol{x}}_0$ is generated as Eq.2 according to the conditions $t$, $\boldsymbol{f}_\kappa$ and $\boldsymbol{f}_\phi$.

**Mesh-specific Motion Learning**. The existing motion-language dataset only involves human motions of SMPL model, which cannot be directly used to learn shape-aware motions for heterogeneous characters. To surmount this constraint, we randomly select characters from the 3D character datasets during training and use a weakly-supervised manner to assist the model in learning the mesh-specific information from the mesh deformation feature extracted by the Mesh Handle Predictor. As the right part of Figure 3 shows, in conjunction with the fully-supervised Motion Loss $\mathcal{L}_m$, we designed two weakly-supervised losses, i.e., Spring Loss $\mathcal{L}_h$ and Adversarial Loss $\mathcal{L}_a$ to achieve mesh-specific motion learning.

The traditional skeletal rigging system of 3D characters has an important property: the length of the bones typically remains consistent during motion. This property ensures that the character's body structure remains stable throughout the animation. Drawing inspiration from this property, we propose the Spring Loss $\mathcal{L}_h$. We model the handles on the character's limbs as a spring system and penalize the length deformation of this system. Specifically, we pre-define the common adjacency relationships between the handles of the character's limbs and use the distance between adjacent handles in the rest-pose as a reference to penalize the distance changes during the character's motion:

$$\mathcal{L}_h = \sum_{i,j \in \mathcal{A}} \sum_{n=1}^{N} \left( e^{-\left(\mathcal{E}(h_i^0, h_j^0) + \sigma\right)} \|\mathcal{E}(h_i^n, h_j^n) - \mathcal{E}(h_i^0, h_j^0)\|_2^2 + \right.$$
$$\left. \|\mathcal{E}(h_i^n, h_j^n) - \mathcal{E}(h_i^{n-1}, h_j^{n-1})\|_2^2 \right), \tag{7}$$

where $\mathcal{E}(\cdot)$ is the Euclidean Distance Function and $\sigma$ is a hyper-parameter. $i$ and $j$ are the indices of the adjacent handles. $h^0$ is the handle position of the rest-pose, $h^n$ is that of the $n$-frame pose. $e^{-\left(\mathcal{E}(h_i^0, h_j^0) + \sigma\right)}$ is an adaptive spring coefficient.

The Adversarial Loss $\mathcal{L}_a$ is inspired by GAN (Goodfellow et al., 2020) and is used to encourage the generated mesh motion to conform to the prior of the skeleton-driven mesh motion on various characters. To reduce computation costs, we randomly sample 100 mesh vertices on the arms in each frame to calculate the Adversarial Loss for each character. This is based on the observation that motion distortion mainly occurs in the arm parts of the characters during movement. We utilize a Discriminator to differentiate the generated handle-driven mesh motions from the skeleton-driven ones. $\mathcal{L}_a$ is designed based on the adversarial training:

$$\mathcal{L}_a = \mathbb{E}_{\bar{V} \sim p(\bar{V}^{sk})} \left[ \log Disc(\bar{V}) \right] +$$
$$\mathbb{E}_{\bar{V} \sim p(\bar{V}^g)} \left[ \log \left( 1 - Disc(\bar{V}) \right) \right], \tag{8}$$

in which $p(\cdot)$ represents the distribution of the skeleton-driven $\bar{V}^{sk}$ or generated $\bar{V}^g$ mesh motions.

With the three losses introduced above, the Shape-aware Motion Diffusion can be trained by:

$$\min_{\boldsymbol{\theta}_\mu} \mathcal{L}_m + \nu_h \mathcal{L}_h + \nu_a \mathcal{L}_a, \tag{9}$$

where $\nu_h$ and $\nu_a$ are the balancing factors. $\mathcal{L}_h$ and $\mathcal{L}_a$ are conducted on the motion adaptation $\hat{\boldsymbol{\delta}}$.

**Motion Adaptation Fine-tuning**. To further enhance the compatibility of the generated motion for heterogeneous characters and minimize the mesh distortion caused by the motion, we utilize pseudo-labels to finetune the linear mapping layer of the motion adaptation. These pseudo-labels are obtained by optimizing the motion adaptation through an as-rigid-as-possible (ARAP) objective defined as:

$$\mathcal{O} = \nu_v \sum_{i,j \in \Phi} \|\mathcal{E}(\bar{V}_i^n, \bar{V}_j^n) - \mathcal{E}(\bar{V}_i^0, \bar{V}_j^0)\|_2^2 + \|\boldsymbol{\delta}' - \hat{\boldsymbol{\delta}}\|_2^2. \tag{10}$$

Here, $\Phi$ represents the set of edges on the mesh, and $\boldsymbol{\delta}'$ represents the optimized motion adaptation at the previous step. $\nu_v$ is the balancing factor.

The linear mapping layer of the motion adaptation is then finetuned by:

$$\min_{\boldsymbol{\theta}_\mu^\delta} \|\boldsymbol{\delta}^{pl} - \hat{\boldsymbol{\delta}}\|_2^2, \tag{11}$$

where $\boldsymbol{\delta}^{pl}$ is the pseudo-labels of the motion adaptations, $\boldsymbol{\theta}_\mu^\delta$ is the learnable parameters.

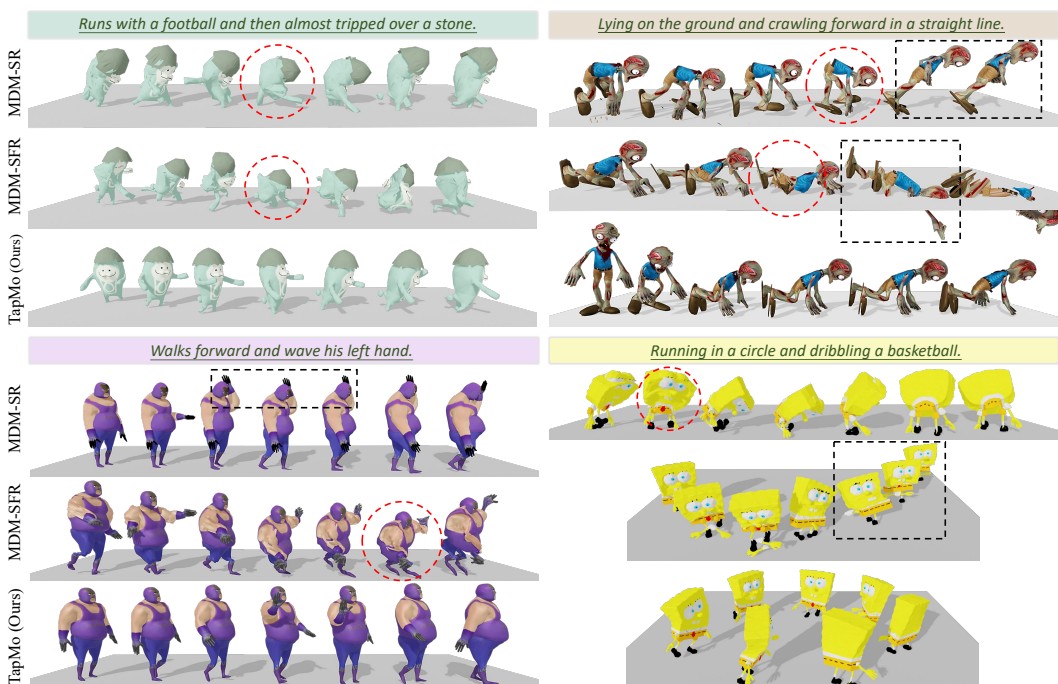

Figure 4: **Qualitative comparison of the generated animations.** The red circle indicates that this sample suffers from severe mesh distortion. The black wireframe indicates that this sample does not align with the motion description.

## 4 EXPERIMENTS

### 4.1 EXPERIMENTAL SETUP

**Baselines**. To the best of our knowledge, TapMo is the first pipeline to generate animations for skeleton-free characters in a text-driven manner. We design two reasonable baselines to evaluate the effectiveness of our TapMo. 1) **MDM + skeleton-based retargeting (MDM-SR).** MDM (Tevet et al., 2022) is a popular motion generation method for the SMPL model. We use MDM to generate human motion based on the text, then automatically rig the skeleton template of the SMPL model to the character by Baran & Popović (2007), and finally retarget the human motion to the character by motion copy. 2) **MDM + SfPT (MDM-SFR).** We use MDM to generate human motion, then use SfPT (Liao et al., 2022) to transfer the human motion to the character frame by frame.

**Datasets**. Four public datasets are used to train and evaluate our TapMo, i.e., AMASS (Mahmood et al., 2019), Mixamo (Adobe), ModelsResource-RigNet (Xu et al., 2019), and HumanML3D (Guo et al., 2022). Please see the Appendix for more details about datasets.

**Metrics**. We quantitatively evaluate our TapMo from two aspects, *i.e.*, motion quality and geometry quality. For evaluating the motion quality, we follow the approach presented in Guo et al. (2022), where motion representations and text descriptions are first embedded using a pre-trained feature extractor and then evaluated using five metrics. However, as the motion representation in our TapMo differs from that in Guo et al. (2022), we adopt the same configurations as theirs and retrain the feature extractor to align with our motion representation. For the geometry quality, we evaluate it from mesh vertices and mesh handles using two metrics as follows: 1) **ARAP-Loss** is used to gauge the level of local distortion of the mesh during the characters' motion, which can be calculated by:

$$\mathcal{L}_{ARAP} = \sum_{i,j \in \Phi} \|\mathcal{E}(\bar{V}_i^n, \bar{V}_j^n) - \mathcal{E}(\bar{V}_i^0, \bar{V}_j^0)\|_2^2, \tag{12}$$

where $\Phi$ is the set of edges on the mesh, ARAP-Loss is similar to the ARAP objective function proposed in Eq.10, but without the second objective item. 2) **Handle-FID**. To assess the global geometry quality, we employ two SMPL human models that vary significantly in body size by adjusting the shape parameters. Using FID, we measure the distribution distances between the handle positions of the generated motion and the real motion on these two models.

We would encourage the reviewers to see more about the implementation details in the Appendix.

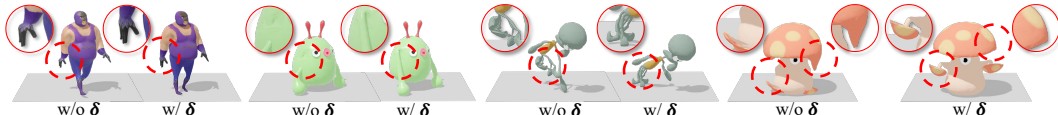

Figure 5: **Visualization of the mesh handles and their corresponding skinning weights.** The skinning weights that correspond to the root handle are highlighted in red.

Figure 6: **Illustration of the impact of the mesh-specific motion in TapMo.** "w/ $\delta$" and "w/o $\delta$" means the animation results with or without equipping the motion adaptations. The mesh-specific motion adaptations generated by TapMo can effectively alleviate the mesh distortion in animations.

## 4.2 QUALITATIVE RESULTS

Figure 4 illustrates the overall qualitative results of our TapMo pipeline, demonstrating its impressive capabilities in generating reasonable motions and animating a diverse range of 3D characters based on textual descriptions. In comparison to baseline models, TapMo's motions more closely align with textual descriptions, exhibit less mesh distortion, and offer higher-quality animation. For instance, as highlighted by the red circle in Figure 4, the results produced by MDM-SR and MDM-SFR demonstrate significant mesh distortion, while TapMo preserves the geometry details better. Additionally, the black wireframe displayed in Figure 4 demonstrates that baseline methods tend to generate motions that do not align with the description, while our TapMo benefits from improved Diffusion Model structure, resulting in generated motion that is more consistent with the description.

Figure 5 provides a visual representation of the adaptive mesh handles and their corresponding skinning weights for various characters estimated by TapMo's Mesh Handle Predictor. Handles not assigned to any mesh vertices are not drawn in this figure. In humanoid characters, the Mesh Handle Predictor accurately assigns handles to the positions that are near the skeleton joints of the human body, resulting in natural and realistic humanoid movements. For non-humanoid characters, the Mesh Handle Predictor assigns mesh vertices to different numbers of handles based on the semantics of the mesh parts. It is worth noting that all characters have a root handle located near the center of their bodies, which serves as the key to driving global movement.

Figure 6 illustrates how mesh-specific motion adaptations impact the animation results in TapMo. Due to the significant variation in the shape and topology of characters, the original motion generated by the Diffusion Model may lead to severe mesh distortion. The results exhibit local mesh distortion and inter-penetrations without motion adaptations. The mesh-specific motion adaptations generated by TapMo can effectively alleviate these issues and enhance the geometry quality.

## 4.3 QUANTITATIVE RESULTS

Table 1 provides a quantitative comparison between TapMo and previous motion generation models, including T2G (Bhattacharya et al., 2021), Hier (Ghosh et al., 2021), TEMOS (Petrovich et al., 2022), T2M (Guo et al., 2022), MDM (Tevet et al., 2022), and ReMoDiffuse (Zhang et al., 2023). The results demonstrate that our TapMo outperforms previous models significantly in terms of both R-Precision and FID metrics. We introduced multi-token and Transformer Decoder strategies to the Shape-aware Motion Diffusion, which showed a stable improvement in all metrics. Notably, TapMo (Ours) achieved a remarkable improvement of over 50% in the FID metric compared to the TapMo (MDM) implementation, with a score of 0.515 compared to 1.058. These findings demonstrate the superior performance of our TapMo and the effectiveness of the adaptations we made to the Diffusion Model structure, which allowed for more accurate and higher-quality motion generation.

Table 2 provides a comprehensive comparison between TapMo and the baseline methods, focusing on the geometry quality of the generated animations. TapMo demonstrates remarkable improvements, achieving a significant reduction in ARAP-Loss both on seen and unseen characters. Specifically, TapMo achieves 69.8% (0.271 vs. 0.897) lower scores compared to MDM-SR and 73.1% (0.271 vs. 1.006) lower scores compared to MDM-SFR on seen characters. For unseen characters, TapMo achieves 58.2% (0.329 vs. 0.787) lower scores compared to MDM-SR and 63.8% (0.329 vs. 0.910) lower scores compared to MDM-SFR. These results indicate that TapMo excels in generating animations with superior geometry details and smoother mesh surfaces compared to the

Table 1: **Comparison of text-based motion generation on the HumanML3D dataset.** "Real†" is the real distribution that is extracted from our motion representation by the retrained feature extractor. "TapMo (MDM)" is a implementation that employs the same diffusion structure as MDM.

| Method | R-Prec. (Top-1)$_\uparrow$ | R-Prec. (Top-3)$_\uparrow$ | FID$_\downarrow$ | Multimodal Dist$_\downarrow$ | Diversity$_\rightarrow$ | Multimodality$_\uparrow$ |
|---|---|---|---|---|---|---|
| Real | $0.511^{\pm.003}$ | $0.797^{\pm.002}$ | $0.002^{\pm.000}$ | $2.974^{\pm.008}$ | $9.503^{\pm.065}$ | - |
| T2G | $0.165^{\pm.001}$ | $0.345^{\pm.002}$ | $7.664^{\pm.030}$ | $6.030^{\pm.018}$ | $6.409^{\pm.071}$ | - |
| Hier | $0.301^{\pm.002}$ | $0.552^{\pm.004}$ | $6.532^{\pm.024}$ | $5.012^{\pm.018}$ | $8.332^{\pm.042}$ | - |
| TEMOS | $0.424^{\pm.002}$ | $0.722^{\pm.002}$ | $3.734^{\pm.028}$ | $3.703^{\pm.008}$ | $8.973^{\pm.071}$ | $0.368^{\pm.018}$ |
| T2M | $0.457^{\pm.002}$ | $0.740^{\pm.003}$ | $1.067^{\pm.002}$ | $3.340^{\pm.008}$ | $9.188^{\pm.002}$ | $2.090^{\pm.083}$ |
| MotionDiffuse | $0.491^{\pm.001}$ | $0.782^{\pm.001}$ | $0.630^{\pm.001}$ | $3.113^{\pm.001}$ | $9.410^{\pm.049}$ | $1.553^{\pm.083}$ |
| MDM | $0.320^{\pm.002}$ | $0.611^{\pm.007}$ | $0.544^{\pm.044}$ | $5.566^{\pm.027}$ | $\mathbf{9.559}^{\pm.086}$ | $\mathbf{2.799}^{\pm.072}$ |
| ReMoDiffuse | $0.510^{\pm.005}$ | $0.795^{\pm.004}$ | $\mathbf{0.103}^{\pm.004}$ | $\mathbf{2.974}^{\pm.016}$ | $9.018^{\pm.075}$ | $1.795^{\pm.043}$ |
| Real† | $0.559^{\pm.007}$ | $0.828^{\pm.005}$ | $0.002^{\pm.001}$ | $4.371^{\pm.035}$ | $17.176^{\pm.178}$ | - |
| TapMo (MDM) | $0.498^{\pm.006}$ | $0.801^{\pm.003}$ | $1.058^{\pm.056}$ | $5.126^{\pm.069}$ | $16.553^{\pm.315}$ | $2.744^{\pm.156}$ |
| TapMo (Ours) | $\mathbf{0.542}^{\pm.008}$ | $\mathbf{0.805}^{\pm.002}$ | $\underline{0.515}^{\pm.088}$ | $5.058^{\pm.063}$ | $16.724^{\pm.380}$ | $2.559^{\pm.230}$ |

Table 2: **Comparison of mesh distortion.** "w/o $\delta$ ft" means the results without adaptation fine-tuning. "A-L" is the ARAP-Loss. "S" and "U" represent the seen and unseen characters.

| Methods | A-L (S)$_\downarrow$ | A-L (U)$_\downarrow$ | Handle-FID$_\downarrow$ |
|---|---|---|---|
| MDM-SR | 0.897 | 0.787 | **0.012** |
| MDM-SFR | 1.006 | 0.910 | 0.163 |
| TapMo (w/o $\delta$) | 0.310 | 0.356 | 0.055 |
| TapMo (w/ $\mathcal{L}_h$) | 0.303 | 0.347 | 0.048 |
| TapMo (w/ $\mathcal{L}_a$) | 0.307 | 0.342 | 0.056 |
| TapMo (w/o $\delta$ ft) | 0.301 | 0.335 | 0.048 |
| TapMo (Ours) | **0.271** | **0.329** | 0.046 |

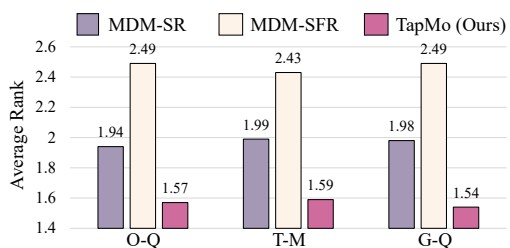

Table 3: **User Study of the animation quality.** We show the average ranking results where the first is the best.

baseline methods. Moreover, TapMo exhibits a superior global geometry preservation performance, with Handle-FID scores 71.7% (0.046 vs 0.163) lower than that of MDM-SFR. Notably, MDM-SR achieves the best Handle-FID (0.012) due to its utilization of a standard SMPL skeleton structure, resulting in minimal global distortion when driving the SMPL model. Additionally, the ablative study underscores the effectiveness of the special designs employed in our TapMo pipeline.

## 4.4 USER STUDY

We conducted a user study to evaluate the visual effects of our TapMo against the baseline methods. We invited 100 volunteers and gave them 15 videos. Each video includes one textual description and three anonymous animation results. We ask users to rank the three results in three aspects: overall quality (O-Q), text-motion matching degree (T-M), and geometry quality (G-Q). After excluding abnormal questionnaires, we collect 1,335 ranking comparison statistics in total, and the average rank of the methods is summarized in Figure 3. Our TapMo outperforms the baseline methods by a large margin and more than 84% of users prefer the animation generated by our TapMo.

## 5 CONCLUSION

In this work, we propose a novel text-driven animation pipeline called TapMo, which enables non-experts to create their own animations without requiring professional knowledge. In TapMo, two key components are exploited to control and animate a wide range of skeleton-free characters. The first is the Mesh Handle Predictor, which clusters mesh vertices into adaptive handles for semantic control. The second is Shape-aware Motion Diffusion, which generates text-guided motions considering the specific deformation properties of the mesh, ensuring coherent and plausible character animation without introducing mesh distortion. To train TapMo with limited ground-truth data of both handle-annotated 3D characters and text-relevant motions, we propose a weakly-supervised training strategy for adaptive mesh handle learning and mesh-specific motion learning. We conduct extensive experiments to validate the effectiveness of our method and show that it achieves state-of-the-art performance compared to baseline methods.

**Acknowledgements.** This work was supported by the fund of Tencent AI Lab RBFR2022012, and the Natural Science Fund for Distinguished Young Scholars of Hubei Province under Grant 2022CFA075.

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

# A APPENDIX

## A.1 SUPPLEMENTARY ON METHOD

We introduce three losses, i.e., Skinning Loss $\mathcal{L}_s$, Root Loss $\mathcal{L}_r$, and Pose Loss $\mathcal{L}_p$ to achieve an adaptive handle learning for our Handle Predictor.

Following Liao et al. (2022), the assumption of $\mathcal{L}_s$ is that if two mesh vertices belong to the same body part based on the ground-truth skinning weight, they should also be controlled by the same handle in the predicted skinning. We select mesh vertices with $s_{i,k} > 0.9, \exists k$ and use the KL divergence to enforce similarity between skinning weights of these two vertices:

$$\mathcal{L}_s = \gamma_{i,j} \sum_{k=1}^{K} \left( s_{i,k} \log(s_{i,k}) - s_{i,k} \log(s_{j,k}) \right), \tag{13}$$

where $i$ and $j$ indicate two randomly sampled vertices. $\gamma$ is an indicator function defined as follows: $\gamma_{i,j} = 1$ if $i$ and $j$ belong to the same part in the ground-truth skinning weight and $\gamma_{i,j} = -1$ if not.

Pose Loss $\mathcal{L}_p$ is to ensure that the posed mesh driven by the predicted handle and skinning weight can be as similar as possible to the shape of the mesh driven by the ground-truth skeleton, which helps the predicted handle more in line with the character's articulation structure. To achieve this, we randomly select a skeletal pose for the character and drive the mesh using its skeleton to obtain a posed mesh $\bar{V}^p$. Then, the local translation and local rotation of our representation can be analytically calculated according to the rest-pose mesh and the posed mesh (Besl & McKay, 1992). Next, we apply the local motion to the mesh using Eq.3 without the global items to obtain a skeleton-free posed mesh $\hat{V}^p$. The $\mathcal{L}_p$ can be calculated by:

$$\mathcal{L}_p = \|\hat{V}^p - \bar{V}^p\|_2^2. \tag{14}$$

A.2 SUPPLEMENTARY ON EXPERIMENTAL SETUP

**Implementation Details**. We implement our pipeline using PyTorch framework(Paszke et al., 2019). The Mesh Handle Predictor (refer to Section 3.1) consists of a three-layer GCN followed by a three-layer MLP with a $softmax$ activation function. The input feature dimension of the Handle Predictor is $V \times 6$, and the output dimension of the skinning weight is $V \times K$. The handle positions, with a dimension of $K \times 3$, are calculated by averaging the positions of vertices weighted by the skinning weight. The number of the mesh handle $K$ is set as 30. The architecture of the Diffusion Model in our TapMo (see Section 3.2) is illustrated in the left part of Figure 3, with the number of attention heads set to 4. The mesh deformation feature $\boldsymbol{f}_\phi$ is formed by concatenating three-level features within the Mesh Handle Predictor, each with dimensions of 64, 128, and 256 respectively. Subsequently, a linear layer is employed in the Diffusion Model to map the mesh deformation feature to a dimension of 512.

In training, the motion sequence length $N$ is padded to 196 frames, and the text length is padded to 30 words. The text feature and the mesh feature are concatenated to form the condition token, with a dimension of $31 \times 512$. Padding masks for the motion sequence and text are utilized during training to prevent mode collapse. The loss balancing factors $\nu_r$, $\nu_p$, $\nu_h$, and $\nu_a$ are set as 0.1, 1.0, 0.001, and 0.1, respectively. We use an Adam optimizer with a learning rate of $1e$-4 to train the Handle Predictor. The batch size is 4, and the training epoch is 500. To train the Diffusion Model, we also use an Adam optimizer with a learning rate of $1e$-4. The batch size is 32 and the training step is 800,000. The finetune step is set as 100,000. The optimization process involves simultaneously optimizing the Diffusion Model and the mesh motion discriminator. In addition, a margin parameter of 0.3 is introduced, dictating that the discriminator is optimized only when the score of the fake sample exceeds the margin parameter, and vice versa.

**Datasets**. Four public datasets are used to train and evaluate our TapMo, i.e., AMASS (Mahmood et al., 2019), Mixamo (Adobe), ModelsResource-RigNet (Xu et al., 2019), and HumanML3D (Guo et al., 2022).

- AMASS is a comprehensive human motion dataset comprising over 11,000 motion sequences. This dataset was created by harmonizing diverse motion capture data using a standardized parameterization of the SMPL model. The SMPL model disentangles and parameterizes human pose and shape, allowing for the generation of a range of body shapes by manipulating the shape parameters.

- Mixamo is an animation repository consisting of multiple 3D virtual characters with diverse skeletons and shapes. In our training process, we utilize 100 characters and 1,112 motion sequences from this dataset. By matching the corresponding joints of different characters, these motion sequences can be applied to any character in the dataset, allowing for a wide range of animations to be used in our pipeline.

- ModelsResource-RigNet is a 3D character dataset that contains 2,703 rigged characters with a large shape variety. Each character has one mesh in the rest pose and has its individual skeletal rigging system with skinning weight. The meshes of these characters are heterogeneous in number, order, and topology of the vertices. We follow the train-test split protocol in Xu et al. (2019).

- HumanML3D is a large-scale motion-language dataset that textually re-annotating motion capture data from the AMASS and HumanAct12 (Guo et al., 2020) collections. It contains 14,616 motion sequences annotated by 44,970 textual descriptions. The motion sequences in HumanML3D are all fit for the SMPL model and represented by a concatenation of root velocity, joint positions, joint velocities, joint rotations, and the foot contact binary labels. We convert these SMPL motions to our ground-truth motions using the analytical method introduced in Besl & McKay (1992) and follow the train-test split protocol in Guo et al. (2020).

**Metrics**. The five metrics (Guo et al., 2022) for evaluating motion quality are listed as follows:

- R-Precision. We provide a single motion sequence and 32 text descriptions, consisting of one ground truth and 31 randomly selected mismatched descriptions. We calculate the Euclidean distances between the motion and text embeddings and rank them. We report the accuracy of motion-to-text retrieval at Top-1, Top-2, and Top-3 accuracy.

- Frechet Inception Distance (FID). Calculating the distribution distance between the generated and real motion using FID (Heusel et al., 2017) on the extracted motion features.

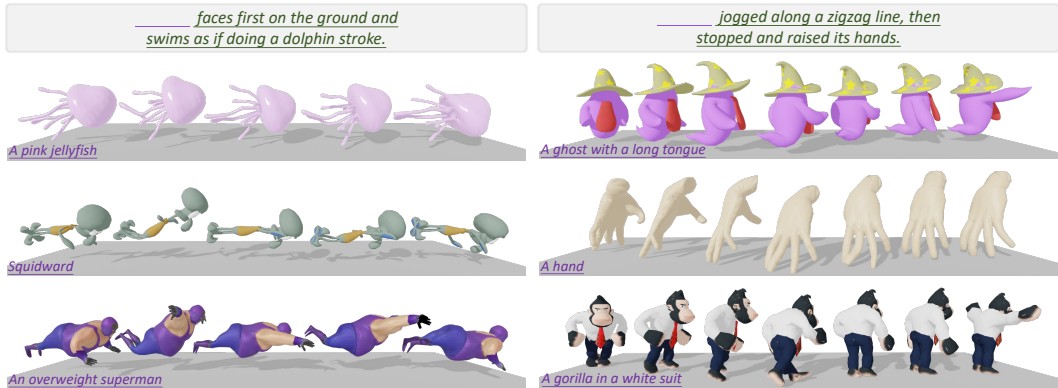

Figure 7: **Qualitative results of executing the same motion descriptions with characters of different shapes.** TapMo can generate mesh-specific motions for characters without causing noticeable mesh distortion while utilizing adaptive handles to drive the characters naturally.

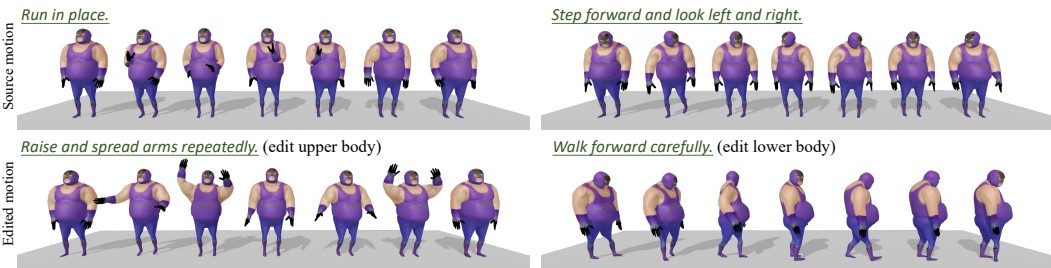

Figure 8: **Examples of animation editing.** The left part and the right part of this figure are the generated results of editing the upper body and lower body of the character, respectively. Please note that we edit the upper body (the left one) and the lower body (the right one) based on different text inputs.

- Multimodal Distance (MM-Dist). The average Euclidean distances between each text feature and the generated motion feature corresponding to this text.

- Diversity. We randomly select 300 motion pairs from a set, extract their features, and calculate the average Euclidean distances between the pairs to quantify motion diversity within the set.

- Multimodality (MModality). We generate 20 motion sequences for each text description, resulting in 10 pairs of motion. For each pair, we extract motion features and calculate the average Euclidean distance between them. The reported metric is the average distance over all text descriptions.

### A.3 SUPPLEMENTARY ON EXPERIMENTS

Figure 7 shows the results of TapMo in generating animations that match the same motion descriptions for characters with vastly different body shapes. TapMo can generate mesh-specific motions for characters without causing noticeable mesh distortion while utilizing adaptive handles to drive the characters naturally. For instance, as depicted in the left portion of Figure 7, TapMo accurately replicates the dynamic postures of both jellyfish and humans swimming. In the example on the right side of Figure 7, a ghost with no legs, a hand, as well as a gorilla, can all execute the running and raising hands motion. Overall, the motion generated by TapMo is plausible and lifelike.

We have implemented an additional application of TapMo, which is animation editing. This application allows users to make edits to the animation during the motion generation process without requiring any additional training. Users can fix handles that they do not want to edit and let the model generate the rest of the animation. We experiment with editing the upper and lower body motion of the characters, which is achieved by overwriting $\hat{x}_0$ with a part of the input motion during the sampling process of the Diffusion Model. Figure 8 demonstrates that TapMo is capable of editing animations according to the text description while maintaining motion consistency.

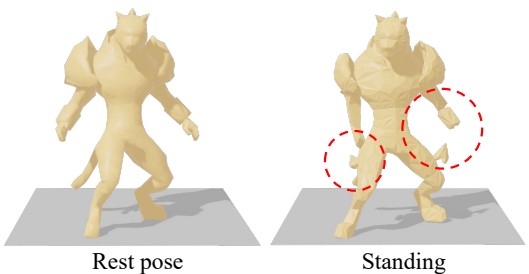

| Rest pose | Standing |

Figure 9: **A failure case.** The generated animation exhibits obvious mesh distortion and breakage when the rest pose of the character is non-standard.

Figure 9 demonstrates a failure case of TapMo, where the character animation exhibits significant mesh distortion and breakage. This issue arises due to the non-standard rest pose of the character, which deviates from the typical T-pose used in our training data. Consequently, TapMo faces limitations in generating animations for characters with non-standard rest poses. It is important to note that this bias related to different rest poses is not unique to our method, as traditional animation creation also requires consistent rest poses for target characters. Addressing this bias and finding solutions for character rest pose variations is an essential area for further research in the field of learning-based animation.

**Generalizability**. TapMo achieves high-quality geometry animations for unseen characters, as indicated by the quantitative results in Table 2. Moreover, the green photo elf and the Sponge Bob in Figure 4, the polygon tree in Figure 5, and the hand in Figure 7 are all internet-sourced wild characters. Most of the characters shown in experiments lacked ground-truth motion data during training. Thus, our TapMo exhibits strong generalizability across diverse mesh topologies, consistently delivering superior-quality animations for both seen and unseen 3D characters.

**Limitations** of this work are listed as follows:

- Due to the weakly-supervised training strategy on SMPL motions, our TapMo is limited in generating rich animations for quadrupeds or characters with significantly different locomotion patterns from humans. A possible solution to this issue is to collect a larger and more diverse dataset of language-motion pairs for a wide range of characters.

- Before the motion generation of TapMo, generating meshes from textual descriptions using existing mesh generation methods like DreamFusion (Poole et al., 2022) or Magic3D (Lin et al., 2022) is a preferable approach to achieve a comprehensive text-based animation pipeline. Unfortunately, current mesh generation methods are constrained to producing simple objects like tables, chairs, or airplanes due to the lack of a character-language dataset. Generating complex animated characters from textual descriptions remains an exciting area for future research.

