# OpenReview forum: "TapMo: Shape-aware Motion Generation of Skeleton-free Characters"
_ICLR.cc/2024/Conference — ICLR 2024 poster_

### Official Review · Reviewer_qE61 · 2023-10-28

**Soundness:** 4 excellent
**Presentation:** 4 excellent
**Contribution:** 3 good
**Rating:** 8
**Confidence:** 4

**Summary:**

The paper (TapMo) tackles text-guided character motion generation in a skeleton-free manner. It can be thought of as the intersection of  MDM (a human-specific motion generation method) and SfPT (a category-agnostic per-frame pose transfer method). Combining the two enables new capabilities -- text-guided motion generation for generic shapes.

Method-wise, instead of simply combining the two prior works, TapMo made innovations on (1) modeling root movement by an additional root handle (2) taking shape into account when generating handle movements (3) introducing a delta term to account for motion that cannot be explained by a fixed number of handles.

The video nicely demonstrates the capability of motion generation beyond human characters.

**Strengths:**

**Significance**
- The paper is a nice first step toward motion generation for generic characters.

**Quality**
- The method is sound and relatively simple.
- The motion generation results are impressive, especially for the walking hand and animals.

**Presentation**
- The visual illustrations are well done, and the paper is easy to follow. I enjoyed reading them.

**Weaknesses:**

**Data**
- The metrics are reported on a human dataset (HumanML3D), which does not show off the cross-category generation ability of the proposed method. Evaluating the method on a dataset with characters beyond humans would be beneficial, such as deforming things 4D [A].

**Method**
- Driving signal. The driving signals seem to be limited to text in the current form. However, there are motions that cannot be described purely by natural language, such as body gestures, facial expressions, hair movements, etc. This is not necessarily a weakness, as the paper already made the setup clear.
- Representation. To generalize to fine-grained motion, such as cloth deformation and hair, the handle-based deformation with a limited number of handles (K=30) seems not enough.


[A] Li, Yang, et al. "4dcomplete: Non-rigid motion estimation beyond the observable surface." Proceedings of the IEEE/CVF International Conference on Computer Vision. 2021.

**Questions:**

1. The model is trained on text-motion pairs of human motion but seems to be able to generalize to other categories like quadruped animals. This is interesting. Is the text description transferable over bipeds and quadrupeds? Has the model seems quadruped motion before?

2. The methods use both diffusion-based reconstruction loss and GAN loss. In which case is the GAN loss necessary/complementary to diffusion loss? What happens if GAN loss is removed?

3. In Table 1, the TapMo variants have a much higher Diversity metric than top rule methods. Is there an explanation? Additionally, the Multimodal Dist and Multimodality results are not explained in the paper. Explanations on why certain method performs better/worse would be useful.

---

> ### Author Response · Authors · 2023-11-20
> **Responses to Reviewer qE61**
>
> (1) **Data.** Thank you for your advice. The experiment on HumanML3D dataset focus on evaluating the motion quality of our text-based motion generation, which ignores the mesh quality of the animation. In addition, we conduct extensive experiments on Mixamo and ModelsResource-RigNet datasets, featuring a diverse set of non-human characters, to evaluate the mesh animation quality as Table 2 and Figure 4 shows.
>
> More generated results for non-humanoid characters and unseen characters can be seen here: https://1drv.ms/v/s!AiJmM4J25MWVzvwreLZLlS_PWkDeHQ?e=rS32C3
>
> (2) **Driving signal.** Thanks for this insightful comment. Whole-body motion generation, such as body gestures and facial expressions, is indeed a valuable research direction that warrants further study.
>
> (3) **Representation.** Yes, the handle-based deformation strategy currently faces challenges in achieving fine-grained motion control, leaving skeleton-based animation methods dominant in this field. However, the handle-based deformation strategy employed in TapMo offers advantages in efficiently controlling a diverse range of characters in a unified and flexible manner. Our qualitative results illustrate that a limited number of handles (K=30) for generating body animations for these characters is sufficient. Additionally, we conducted tests using 60 handles, the same as SfTP. However, the animation results were inferior to using 30 handles, mainly due to an excess of redundant handles. This abundance adversely affects the Handle Predictor's ability to learn reasonable skinning weights, impacting the overall animation quality.
>
> (4) **Quadruped animals.** TapMo proves its versatility by successfully animating non-humanoid upright characters like the slime in Figure 1, the potato elf in Figure 4, and the jellyfish, the ghost in Figure 7, enabling them to move in a human-like manner. Additionally, TapMo can drive animals with multiple feet to perform simple actions like walking or running, treating their bodies as human bodies without arms. These quadruped animals and motions haven’t seen by the model in training. These results convincingly demonstrate the remarkable adaptability of our TapMo to characters with diverse shapes and forms. When it comes to more difficult motion of quadrupeds, such as catching prey, their distinct movement patterns significantly differ from those of humans, making it impossible to learn them solely from human motion data. Nevertheless, leveraging the versatility of our Handle-based Mesh Deformation approach, our TapMo system can be extended to animate quadrupeds by training it on quadruped mocap data.
>
> (5) **GAN loss.** As indicated by the results presented in Table 2, the inclusion of the GAN loss proves beneficial for our shape-aware motion diffusion model in generalizing to unseen characters. This is evident in the lower "A-L(U)" metric achieved when incorporating the GAN loss. However, the GAN loss will also cause a slight increase in the "A-L(S)" and Handle-FID metrics. Since generalization ability is our focus in dealing with a wide variety of characters, we use the GAN loss in our TapMo.
>
> (6) **Metrics in Table 1.** The value of Diversity metric depends heavily on the pre-trained motion feature extractor. Since the motion representation in our TapMo is differ from that in standard HumanML3D, we adopt the same configurations as theirs and retrain the feature extractor to align with our motion representation. When compared with our real distribution, the Diversity metric of TapMo decreases by 0.452 (16.724 vs. 17.176). This aligns with the trends observed in existing methods, such as T2M and TEMOS compared with their real distribution. The Diversity metric of MDM is most closely aligned with its real distribution (9.559 vs. 9.503).
>
> The analysis method for the Multimodal Dist metric is similar to Diversity. For Multimodal Dist, our TapMo is superior to MDM with 0.687 higher than our real distribution, while MDM is 2.592 higher than its real distribution. T2M demonstrates the best performance on Multimodal Dist with 0.366 higher than its real distribution. As for Multimodality, TapMo performs at a comparable level with MDM and significantly outshines other methods.

---

### Official Review · Reviewer_e3Gp · 2023-10-30

**Soundness:** 3 good
**Presentation:** 3 good
**Contribution:** 2 fair
**Rating:** 6
**Confidence:** 4

**Summary:**

This work focuses on an interesting research topic - synthesizing motions for skeleton-free 3D characters, with two main modules: 1. Mesh Handle Predictor, and 2. Shape-aware Motion Diffusion Module.  In addition, this work utilizes the shape deformation-aware features as a condition to guide the motion generation for specific character models. The proposed method could show impressive generated animations for both seen and unseen characters.

**Strengths:**

1. it is good to study generating shape-aware motions, especially for non-humanoid 3D characters.
2. The proposed method seems to be reasonable and might be promising to generate motions for unseen characters.

**Weaknesses:**

1. The proposed mesh handle predictor is simple and straightforward, but it is not clear how the proposed method resolves different characters that have different topologies with different semantics.  Currently, the manuscript mentions that "each handle is dynamically assigned to vertices with the same semantics across different meshes", but it is not clear how the method will select those handles.
Also, it is unclear how the method will choose the number of handles since different topologies tend to have different numbers of handles. It is unclear how the proposed method could achieve training with different numbers of handles.

2. The proposed Shape-aware Motion Diffusion seems to be simple modifications for existing methods (e.g., MDM), but the current presentation makes it overcomplicated, and difficult for readers to get the key designs that could highly improve the motion generation quality. I am not sure if considering the character shapes is the major factor that improves the quality, and the others could make further improvements.

3. The experiments seem to be insufficient.  HumanML3D dataset is the only benchmark that is used to report quantitative comparisons.  However, recent methods, such as MotionDiffuse[1], ReMoDiffuse[2], and T2M-GPT[3] have not been discussed.

[1] Zhang, M., Cai, Z., Pan, L., Hong, F., Guo, X., Yang, L., & Liu, Z. (2022). Motiondiffuse: Text-driven human motion generation with diffusion model. arXiv preprint arXiv:2208.15001.

[2] Zhang, M., Guo, X., Pan, L., Cai, Z., Hong, F., Li, H., ... & Liu, Z. (2023). ReMoDiffuse: Retrieval-Augmented Motion Diffusion Model. ICCV 2023

[3] Zhang, J., Zhang, Y., Cun, X., Huang, S., Zhang, Y., Zhao, H., ... & Shen, X. T2M-GPT: Generating Human Motion from Textual Descriptions with Discrete Representations. CVPR 2023

**Questions:**

The authors could answer my questions asked in the weakness section first.
Besides, I would like to suggest the authors could improve its readability and also highlight the key design.
For the experiments, I would like to see more generated motions, especially for non-humanoid characters and unseen characters.

---

> ### Author Response · Authors · 2023-11-20
> **Responses to Reviewer e3Gp**
>
> (1) **The proposed Mesh Handle Predictor** is implemented through a GCN-based network capable of handling graph structures with an arbitrary number of vertices and topologies. This enables the processing of different characters with varying numbers of mesh vertices and distinct mesh topologies in a unified manner.
>
> We predefine K=30 handles for all characters and utilize the Handle Predictor to learn and predict a V*K-dimensional skinning weight, assigning the V mesh vertices to K handles. The vertex number V is specialized for different characters. With the exception of the first handle, designated as the root handle, the network autonomously learns the semantics and order of the mesh handles. For instance, in the case of a character lacking legs, as illustrated by the mushroom in Figure 5, vertices will not be allocated to handles associated with leg semantics. In other words, all vertices will have approximately zero skinning weight for the leg handles.
>
> During training, our Mesh Handle Predictor is trained without handle annotations. It learns the semantics of mesh topologies in a weakly-supervised manner by optimizing the loss in Eq. 5.
>
> (2) **Shape-aware Motion Diffusion.** The central aim of our Shape-aware Motion Diffusion is to generate motion for a diverse range of skeleton-free characters. The primary challenges lie in the inherent variation in shapes and mesh topologies among different characters, and employing a uniform mesh deformation method can readily result in mesh distortion. Therefore, the consideration of character shapes stands out as the key factor in enhancing the quality of the generated animations.
>
> While the latest diffusion-based motion generation methods have proposed significant enhancements to improve the quality and diversity of generated motion, the focus of our work differs significantly. Our primary objective is not aligned with those methods. Instead, we concentrate on animating a diverse array of non-rigged characters uniformly, placing emphasis on ensuring the quality of animation across this variety.
>
> (3) **Experiments.** Thanks for pointing out this problem. In the updated version, we have added citations and provided comparisons with these two diffusion-based methods that are relevant to our work.
>
> (4) Thanks for your advice. We will revise our manuscript carefully to highlight our contribution and introduce our method more clearly. More generated results for non-humanoid characters and unseen characters can be seen here: https://1drv.ms/v/s!AiJmM4J25MWVzvwreLZLlS_PWkDeHQ?e=rS32C3

---

> ### Comment · Reviewer_e3Gp · 2023-11-22
> **A few more questions**
>
> Thanks for the response, and I have a few more questions to discuss with the authors.
>
> 1. I am not sure whether those two weak supervisions (i.e.,$L_h$ and $L_a$) really work. It seems this manuscript does not mention where these loss terms were added. Assuming they were added on the predicted x0, the noise should also be quite significant. Maybe the authors could explain more.
>
> 2. Moreover, the consistency loss for skeletal lengths itself is not strong; in fact, achieving consistent skeletal lengths could be accomplished through forward kinematics.  The reported results from baseline methods using forward kinematics are very bad, which however are expected to be better.  Hence, I'm not sure if there might be an issue with the implementation.

---

> > ### Author Response · Authors · 2023-11-23
> > **Responses to Reviewer e3Gp**
> >
> > (1) Thank you for bringing this issue to our attention. These two weak supervisions are conducted on the Motion Adaptation, which is not directly added on the predicted x0 using stop gradient. The quantitative results in Table 2 ablate these two losses to verify their effectiveness.
> >
> > We have added a detailed description of these two losses in the revised paper (Section 3.2, page 6.).
> >
> > (2) The results of MDM-SR, which involves rigging the SMPL skeleton to various characters and using forward kinematics to drive the characters, are actually unsatisfactory. The main reason is that the SMPL skeleton is not suitable for different characters whose shapes vary from the typical SMPL human, as illustrated by the Photo Elf and SpongeBob in Figure 4. In other words, it is the skeleton mismatch that causes animation distortion rather than the consistency of bone lengths. Consistent bone lengths can reduce distortion, as animated characters typically follow the motion rules of an articulated structure.

---

### Official Review · Reviewer_JBvr · 2023-11-01

**Soundness:** 3 good
**Presentation:** 3 good
**Contribution:** 3 good
**Rating:** 6
**Confidence:** 4

**Summary:**

TapMo is a text driven motion synthesis framework for skeleton-free 3D characters. Addressing limitations of relying on pre-rigged character models, TapMo introduces the Mesh Handle Predictor and the Shape-aware Motion Diffusion. These components enable the framework to generate motions of skeleton-free characters using text descriptions. Specifically, Mesh Handle Predictor predict the skinning weights and can clusters mesh vertices into adaptive handles. Then the Shape-aware Motion Diffusion takes mesh deformation feature and output handles' motion. Trained in a weakly-supervised manner, TapMo demonstrates its performance in generating animations for novel non-human characters.

**Strengths:**

The research addresses an interesting and promising problem, as far as I know it is the first attempt to enable text-driven motion synthesis for skeleton-free characters.

Comprehensive experiments are conducted, yielding impressive results across diverse shapes. Supplementary videos and a user study further validate the naturalness of the generated results.

The combination of diffusion-based motion synthesis and skeleton-free mesh deformation is interesting and novel.

**Weaknesses:**

Some details are not clearly explained, such as the mesh deformation feature, what exactly is f_ and how it's obtained, and its dimensions, which are not reflected in the main text. From the appendix, it seems to be a 512-dimensional vector. Further explanation from the authors is desired. And how does mesh-specific adaptation affect the vertices, it is not included in the equations. How is the Discriminator implemented? Are the two modules trained jointly or separately?

What are the visualization and qualitative results for Handle-FID? Specifically, what are the two SMPL human models and what do the results look like? On the other hand, are two models enough? Why don't authors try MGN ("Multi-garment net: Learning to dress 3d people from images") which includes clothed human models and can use SMPL pose parameter to drive.

I recommend the authors discuss the following skeleton-free papers: "Zero-shot Pose Transfer for Unrigged Stylized 3D Characters" and "HMC: Hierarchical Mesh Coarsening for Skeleton-free Motion Retargeting". The self-supervised shape understanding method in the former could potentially strengthen handle prediction including mesh deformation feature extraction.

The authors could try using the PMD metric in SfPT and the previous works, which allows for direct comparison with ground truth vertices and can judge both motion and mesh quality, this is suitable for SMPL-based models.

Can the authors provide qualitative results before and after Motion Adaptation Fine-tuning? The post-process usually results in big differences in naturalness.

**Questions:**

Compared to Eq1 in SfPT, if we ignore the global translation and rotation of the root, authors added $\tau^{l}_k$ and $h_k$, which seems different from SfPT Eq1. Could the authors explain the purpose of this?

For local translations and local rotations, their first dimension is K. Does this include the root (first) handle? Is k=1 represent root handle in Eq. 3? Or there are K-1 local translations/rotations from k=2 to k=K.

In Eq.7, defining handle adjacent handles is required, is this something that needs to be done separately for each character? For instance, shapes with large differences as in Figure 6. Is this only necessary during training or during inference? And how many characters need to be specifically defined?

In the first paragraph of the Method section, does "skinning weight s of the handle" refer to the skinning weights of the vertices?

---

> ### Author Response · Authors · 2023-11-20
> **Responses to Reviewer JBvr**
>
> (1) **Mesh deformation feature.** We thank you for bringing this issue to our attention. The mesh deformation feature is formed by concatenating three-level features within the Mesh Handle Predictor, each with dimensions of 64, 128, and 256 respectively. Subsequently, a linear layer is employed to map the mesh deformation feature to a dimension of 512. This transformed feature is then concatenated with the text token to construct the condition token in diffusion model. The mesh deformation feature encapsulates both the shape and skinning information of the non-rigged character in latent space, providing a comprehensive representation for further shape-aware motion generation.
>
> The mesh-specific adaptation refers to a local translation vector associated with the predicted mesh handles. The final local translation of the handles is the cumulative result of the generated local translation and the adaptation. Consequently, the mesh vertices can be deformed through Eq. 3.
>
> The implementation of the discriminator is inspired by VIBE (CVPR 2020, Kocabas Muhammed et. al.), incorporating a GRU module to extract motion sequence features and an Attention module to fuse these features. The final classification of the motion sequence is achieved through a linear mapping with a sigmoid function. Notably, before feature fusion, we concatenate the features with the mesh deformation feature to enable the discriminator to perceive the mesh shape as well.
> The discriminator and the diffusion model are trained jointly.
>
> We have included a comprehensive description of the mesh deformation feature in the revised Appendix (Page 13, Implementation Details).
>
> (2) **Handle-FID.** The two SMPL human models and their predicted handles utilized in our experiments can be visualized through the following link (https://1drv.ms/f/s!AiJmM4J25MWVzsMKfQ9Fx2L-RXFg3w?e=0aeCBy). It's best to view the OBJ files in MeshLab so that you can observe the skinning weights. We manipulate the beta parameters in the SMPL model to introduce a significant shape difference between the two models. The color of the mesh surface indicates the predicted skinning weights.
>
> The purpose of introducing the Handle-FID is to evaluate the global geometry quality of the animation results. Thus, for ease of implementation without losing universality, we use beta parameters to adjust the shape of the SMPL model instead of using the MGN.
>
> (3) **Skeleton-free papers.** Thanks for your insightful advice. The ZPT introduces an implicit pose deformation module, which contrasts with our explicit mesh deformation approach. On the other hand, the HMC represents an enhanced version of SfPT, employing a coarse-to-fine transfer of mesh motion to mitigate mesh distortion. However, it's noteworthy that the computational costs associated with HMC are considerably higher than those of SfPT. In our study, we propose the learning of mesh-specific adaptation to address the mesh distortion issue, akin to the fine-grained deformation in HMC but with improved efficiency.
>
> We have incorporated citations and provided a detailed analysis of these two relevant papers in the final version of our paper (Section 2. Related Work. Skeleton-free mesh deformation).
>
> (4) **PMD metric.** Given that TapMo is a generative model producing diverse motion outputs, calculating the PMD metric using the ground-truth mesh motion of the SMPL model becomes impractical.
>
> (5) **The Motion Adaptation Fine-tuning** strategy is helpful for preserving the shape geometry of the characters. The qualitative results can be seen here: https://1drv.ms/v/s!AiJmM4J25MWVzvwqiMJ3f-TJaiyzLg?e=yAm9x8
>
> (6) **Eq1 in SfPT.** The practical implementation of Eq. 1 in the SfPT code aligns with our approach. Although our formula expression differs from Eq. 1 in SfPT, it is because we choose to separately represent the translation term instead of combining it into the rotation matrix. This decision is made to enhance the clarity of expressing the mesh deformation process.
>
> (7) **Local translations and local rotations.** The local translations and local rotations have K-1 dimensions in the first dimension, excluding the root handle. We appreciate you for pointing out this problem. We have thoroughly revised the content to rectify the error and enhance the overall clarity of the writing.
>
> (8) **Adjacent handles.** We predefine the handle adjacency based on three principles: 1) clear semantics of handles, 2) shared by most characters, and 3) significant impact on mesh deformation. As illustrated in Figure 3 ``Spring Loss '', we establish the handles' adjacency on the arms and legs of the character. This approach eliminates the need for separate processing for each character, as all characters share the same handle adjacency. During inference, the handle adjacency is not needed.
>
> (9) **Skinning weights of the vertices.** Yes. Thank you for helping us correct this writing error.

---

### Official Review · Reviewer_6xCW · 2023-11-01

**Soundness:** 3 good
**Presentation:** 3 good
**Contribution:** 3 good
**Rating:** 6
**Confidence:** 4

**Summary:**

The main contribution of this paper is a motion diffusion model that can take shape-deformation features as inputs to generate shape-aware motions. This function is desirable in the computer animation, which can save a lot of efforts in the animation production. The proposed pipeline has two components: mesh handle predictors to predict skinning weights and underlying skeletons and shape-aware motion diffusion models that can synthesizes motion with mesh-specific adaptations.

**Strengths:**

1.  the generated animations for a variety of 3D characters are impressive. The structure of the 3D meshes are well recognized when associating it with the animations.

2. The application of shape deformation feature in animation is nice.

**Weaknesses:**

There are still penetrations between foot and ground in the generated animations, which downgrade the animation quality.

**Questions:**

Please clarify how the adversarial loss is trained in the rebuttal.

---

> ### Author Response · Authors · 2023-11-20
> **Responses to Reviewer 6xCW**
>
> (1) **Foot-ground penetration.** Thank you for the positive feedback and for bringing attention to the issue of foot-ground penetration in animations—a challenge that is indeed widespread in most text-motion generation models. We appreciate the opportunity to discuss this further. In current kinematic-based motion generation models, addressing this problem typically involves post-processing steps. In contrast, PhysDiff [Ye Yuan et.al. ICCV 2023] employs advanced physical simulation techniques to dynamically adjust foot-ground interactions, resulting in a significant reduction in penetrations. This method represents a novel approach in the realm of text-motion generation models. We are committed to exploring and addressing this challenge in our future research endeavors, striving to find a more effective resolution for skeleton-free animations.
>
> (2) **The adversarial loss** is one of our special design for the skeleton-free animation pipeline, which helps the network generate reasonable motion adaption to reduce mesh distortion.
>
> Specifically, a mesh motion discriminator is designed to process the mesh vertices of a motion sequence along with the character's shape feature as inputs. It then produces a classification result indicating whether the mesh motion is driven by the predicted handles or the pre-rigged skeleton. The input mesh vertices are automatically sampled on the character's arms using the predicted handles and skinning weights. Recognizing that skeleton-driven character arms can maintain a reasonable articulated structure and shape in a motion sequence, we employ the adversarial loss to penalize abnormal deformations resulting from the handle-based driving approach.
>
> The adversarial loss is formulated as Eq.8 in the paper. The optimization process involves simultaneously optimizing the diffusion model and the discriminator using adversarial loss and diffusion-based reconstruction loss. In addition, a margin parameter of 0.3 is introduced, dictating that the discriminator is optimized only when the score of the fake sample exceeds the margin parameter, and vice versa.
>
> We have added the detailed training strategy in the revised Appendix (Page 13, Implementation Details).

---

### Author Response · Authors · 2023-11-20
**Responses to all the reviewers**

We would like to express our gratitude to the reviewers for their insightful comments, which have significantly contributed to enhancing the quality of our paper. We have carefully considered all the feedback received, encompassing aspects such as writing, experiments, and analysis. In the revised paper, all modifications have been clearly marked in red.
More generated results for non-humanoid characters and unseen characters can be seen here: https://1drv.ms/v/s!AiJmM4J25MWVzvwreLZLlS_PWkDeHQ?e=rS32C3

---

### Meta-Review · Area_Chair_u2Ww · 2023-12-05

**Metareview:**

This paper proposes a novel task that animate unrigged 3D characters given text prompt describing the target motion. The method includes a) Mesh Handle Predictor that takes an unrigged mesh as input and predicts the handle locations as well we skinning weights, and b) Motion Diffusion model that predicts the motion of the character conditioned on a text prompt and features from the Mesh Handle Predictor. Reviewers agree that this paper proposes an interesting novel task and shows impressive results. Most concerns are around implementation and design details. Reviewers also raised issues such as missing comparisons, ablation studies, citations, which have been included in the revision PDF.

**Justification For Why Not Higher Score:**

Though the task is interesting and the results are impressive, this paper aims at a specific task and has limited impact. Furthermore, most components of the propose methods are already adopted by existing works. Thus I do not recommend the paper for spotlight or oral presentation.

**Justification For Why Not Lower Score:**

All reviewers agree that the paper aims at a novel task and demonstrate impressive results, most concerns regarding implementation details, missing citations, baselines and ablations have been adequately addressed during the rebuttal, thus I recommend the paper for acceptance.

---

### Decision · Program_Chairs · 2024-01-16

Accept (poster)